# Analysis of pH and Electrolytes in Blood and Electrolytes in Ruminal Fluid, including Kidney Function Tests, in Sheep Undergoing General Anaesthesia for Laparotomy

**DOI:** 10.3390/ani12070834

**Published:** 2022-03-25

**Authors:** Lucie Marie Grimm, Martin Ganter

**Affiliations:** Clinic for Swine, Small Ruminants and Forensic Medicine, University of Veterinary Medicine Hannover, Foundation, Bischofsholer Damm 15, 30173 Hannover, Germany; martin.ganter@tiho-hannover.de

**Keywords:** Sectio Caesarea, ovine, blood gas, omasum, oxygen saturation, partial pressure of carbon dioxide, renal function, total intravenous anaesthesia, inhalational anaesthesia

## Abstract

**Simple Summary:**

Caesarean section is a common surgery performed in sheep, wherefore the animal is often placed in the supine position, which can lead to compromised lung function caused by pressure from the rumen and the full uterus. The forestomach system in ruminants contains fluids and electrolytes, which can function as a great reservoir for the blood equilibrium. The ruminal fluid as well as kidney function in sheep undergoing surgical procedures has so far been little examined in research. Therefore, the aim of this study was to evaluate the influence of the type of anaesthesia as well as pregnancy status on these parameters. Based on our results, we can state that inhalational anaesthesia with added oxygen improves the oxygen saturation and partial pressure of oxygen in the blood of sheep in the supine position. Kidney function can be maintained with a minimal electrolyte infusion regime and changes in electrolytes in ruminal fluid could be observed.

**Abstract:**

Background: Performing Sectio Caesarea in sheep under general anaesthesia is a common procedure in veterinary practice. The abdominal cavity can be accessed via linea alba, for which the ewe is positioned in the supine position, whereby rumen and uterus can compromise lung function. Although the rumen represents an important reservoir for fluid and electrolytes, and kidney function during anaesthesia is essential, these parameters have not been focused on in research. Therefore, the objective of this study is to contribute data on blood parameters, ruminal fluid, and kidney function tests during laparotomy. Methods: Laparotomy was performed in 14 ewes, whereof five animals were pregnant ewes (PE) and nine non-pregnant ewes (NPE). A total of seven animals received isoflurane in addition to oxygen (inhalational anaesthesia (InhA)) and seven ewes were anaesthetised with xylazine and ketamine (total intravenous anaesthesia (TIVA)); all ewes received lumbosacral anaesthesia. Blood, urine, and ruminal fluid were sampled every hour over a three-hour period. Results: On comparing InhA to TIVA, higher values were detected for TIVA in haemoglobin, paced cell volume, sodium, phosphate, glucose concentration in the blood, and phosphate in ruminal fluid. Lower values were detected for TIVA in partial pressure of oxygen, oxygen saturation, and creatinine clearance. On comparing PE to NPE, higher values were detected in PE in magnesium and ruminal calcium. Lower values in PE were detected in chloride, base excess in the blood, and ruminal phosphate. Over time, an increase in partial pressure of carbon dioxide, glucose in the blood, glucose in urine, and a decrease in protein and albumin could be observed. Conclusion: Surgery in sheep in the supine position should be performed with additional oxygen to maintain physiological pO_2_ and sO_2_ values. Kidney function could be maintained with a minimal electrolyte infusion regime. Additional glucose is not necessary, even in pregnant ewes. Further research should be conducted on parameters in ruminal fluid.

## 1. Introduction

Surgery in sheep under general anaesthesia for Sectio Caesarea (SC) is a relatively common procedure in veterinary practice, and contrary to cattle [1], the surgery is usually performed under general anaesthesia. The abdominal cavity can be accessed in the left flank or via the ventral mid-line [2]. Various possibilities are described to generate anaesthesia in sheep [3], but bottlenecks regarding the approval of anaesthetics significantly reduce the medicaments of choice. Moreover, in Germany, application of non-steroidal anti-inflammatory drugs (NSAIDs) in sheep require drug repurposing from other species. Voigt et al. found that most ewes are presented for SC due to insufficient cervical dilatation (44.3%) [4]. Maternal survival rate was described as 97.8% under field conditions [4] and in studies comparing elective *SC* to vaginal delivery, postnatal deaths in ewes are similar [5]. In ewes presented at a veterinary teaching hospital, ewe mortality up until hospital discharge was 10.8%, as survival rate is mostly influenced by delayed presentation to a veterinarian [4]. In the following season, previously operated ewes gave birth to fewer live-born lambs, but birth weight and conception rate were not influenced [5]. 

There are different techniques described for SC in sheep or goats, such as accessing the abdominal cavity via the left flank or the linea alba, or, to avoid injury to large veins, slightly paramedian. Access via linea alba is the method taught at the Clinic for Small Ruminants, as it might has a slight advantage in wound healing [2] and is an approach that is also suitable for other surgeries on the abdominal cavity [6,7]. Sheep have a prominent part of the abdominal cavity filled with the rumen (10–15%) [8], which gives them the unique ability to temporarily replace extracellular fluid with fluid from the rumen in situations of limited access to water [9]. In sheep presented for emergency SC, usually no deprivation from food takes place prior to surgery. As a consequence, the veterinarian should consider a full rumen together with a full uterus compressing major blood vessels such as aorta and the lungs when positioning the sheep in supine position. Due to this fact, we suspected a severe influence on blood gas and oxygen saturation. When sheep are operated in the supine position for SC under intravenous anaesthesia, hypoxaemia occurs, and advantages of additionally delivered oxygen during anaesthesia are described [10,11]. Electrolytes in ruminal fluid are so far not well documented during anaesthesia [12], although the rumen is a great reservoir for electrolytes and fluid [13,14]. The rumen generates a remarkable difference to monogastric animals, and its role during anaesthesia has so far not been focussed on. Although kidney function tests have been evaluated in healthy sheep [15], pregnant ewes, ewes with metabolic diseases [16], and sheep with nephropathies [17], only one previous study includes kidney function tests during anaesthesia [12], though the kidney greatly influences electrolytes in the blood [18]. 

The aim of the present study is to contribute data on sheep undergoing surgical procedures in the supine position under two different anaesthesia protocols and to generate data on blood parameters, ruminal fluid, and kidney function tests during this procedure.

## 2. Materials and Methods

Data on 14 female sheep of different breeds (predominantly German blackhead, East Frisian milk sheep and Cameroon sheep) undergoing a laparotomy under general anaesthesia were collected at the Clinic for Swine, Small Ruminants and Forensic Medicine, University of Veterinary Medicine Hannover, Foundation, Germany over a period of two years between May 2017 and January 2019 and evaluated retrospectively. The animals were partly from the university flock or bought from a local livestock dealer and healthy, determined by means of a thorough clinical examination. All animals were operated on in the teaching clinic with students and veterinarians. A total of five animals were pregnant at day 143 *post inseminationem* and nine animals were not pregnant. The surgeries were approved by the Lower Saxony State Office for Consumer Protection and Food Safety (LAVES), Germany under the approval number 33.9-42502-05-09-A627.

### 2.1. Anaesthesia and Pain Management

#### 2.1.1. Setting: Inhalational Anaesthesia (InhA)

A total of seven ewes underwent surgery using inhalational anaesthesia (InhA). As induction, non-pregnant ewes (NPE) n = 3, (mean 60.8 kg bodyweight (BW)) received xylazine (Xylavet 20 mg/mL^®^, CP-Pharma Handelsgesellschaft mbH, Burgdorf, Germany; 0.2 mg/kg BW, i.m.) and ketamine (Ketamin 100 mg/mL^®^, CP-Pharma Handelsgesellschaft mbH, Burgdorf, Germany; 5 mg/kg BW, i.v.), while the pregnant ewes (PE) n = 4 (mean 54.3 kg BW) were induced with ketamine (10mg/kg BW, i.v.) only. All animals were intubated with an endotracheal tube (Endotracheal PVC Tube, 6-7 mm^®^, Kruuse A/S, Langeskov, Denmark). Anaesthesia was then maintained with isoflurane (Forene 100%^®^, Ab-Vie Deutschland GmbH und Co KG, Berlin, Germany) at a concentration of 0.5–2.5% in 50–100% of oxygen using a semi-closed circuit rebreathing system (Modell CATO^®^, Dräger Medizintechnik GmbH, Lübeck, Germany). All ewes were mostly ventilated under artificial respiration with a positive end-expiratory pressure (PEEP) of 5 mbar. Just one PE received xylazine (0.2 mg/kg BW, i.m.) after the lambs had been delivered.

#### 2.1.2. Setting: Total Intravenous Anaesthesia (TIVA)

In seven ewes, surgery was performed using total intravenous anaesthesia (TIVA). NPE n = 6 (mean 39.8 kg BW) received xylazine (0.2 mg/kg BW, i.m.) and ketamine (5 mg/kg BW, i.v., redosed with 2.5 mg/kg BW i.v. every 10–30 min). Just one PE (52 kg BW) was induced with ketamine (10 mg/kg BW, redosed with 5 mg/kg BW i.v. every 10–30 min) and received xylazine (0.2 mg/kg BW, i.m.) after the lambs had been delivered.

All 14 animals received intravenous fluids (full Electrolyte solution: Elektrolyt-Lösung, Albrecht GmbH, Aulendorf, Germany), an intravenous application of NSAIDs, and lumbosacral administration of procaine (Isocain ad us. vet.^®^, Selectavet Dr. Otto Fischer GmbH, Weyarn-Holzolling, Germany; 2–5 mL) 15 min prior to the surgical procedure [19,20], after induction of the general anaesthesia. All PE and one NPE received Amoxicillin (Amoxisel-Trockensubstanz 100 mg/mL^®^, Selectavet Dr. Otto Fischer GmbH, Weyarn-Holzolling, Germany; 10 mg/1 kg BW). For an overview on anaesthesia, see Table 1.

### 2.2. Preparation of the Animal

Fastening of hay and pellets commenced at least 19 h prior to the surgical procedure, without restricted water intake. All ewes had a venous catheter inserted into the jugular vein [21]. After general anaesthesia, an arterial catheter (VasoVet Braunüle G24 0.7 × 19 mm gelb^®^, B. Braun Melsungen AG, Melsungen, Germany) was inserted into the *A. auricularis* in one of the ears. The urinary bladder was catheterised with a Foley catheter (Foley Ballonkatheter, CH 08, 3–5 mL Ballon, Silcoat^®^, Jena, Germany) under visual control. The rumen was accessed by an orally placed feeding tube (Typ Levin 6 mm diameter, 125 cm lenght^®^, Vygon, Écouen, France). All animals were placed in the supine position on a foam rubber mat with the head tilted slightly downwards.

### 2.3. Surgery

The animals were operated on for teaching purposes by 10th semester students under the supervision of one veterinarian per animal. After sterile preparation of the surgical field, the abdominal cavity was opened for about 15 cm in the linea alba caudal to the umbilicus (or slightly paramedian to spare major veins). In PE, the uterus was opened at the curvatura major to extract the lambs and afterwards closed by a double inverting, non-perforating suture. In all animals, the peritoneum and tendon forming the linea alba were closed with Sultan stiches. A subcutaneous suture was applied to reduce wound cavity and the skin was closed intradermally. All sutures were performed with absorbable material (polyglycolic acid).

### 2.4. Sampling

Body weight was measured before surgery. Ruminal fluid was sampled through the inserted oesophageal tube, the urinary catheter was blocked five minutes before sampling, and a urine sample was then taken parallel to a venous and arterial blood sample (S-Monovette 2.6 mL Lithium-Heparin^®^, Sarstedt AG & Co. KG, Nümbrecht, Germany). The rectal temperature was measured manually, and heart rate (HR) and respiratory rate (RR) were determined through manual auscultation. In seven animals, an arterial blood sample was collected two to eight hours after the animals had recovered and were able to stand and walk again.

### 2.5. Analysis of the Samples

The arterial blood sample was analysed within five minutes in the Rapidlab (Rapidlab 1265^®^ Siemens AG, Munich, Germany) (samples of six animals) or Osmetech (Osmetech opti^TM^ CCA, GenMark Diagnostics Europe GmbH, Zug, Switzerland) (samples of eight animals). Both analysers measured (corrected by body temperature) pH (pH_art), partial pressure of carbon dioxide (pCO_2_), and partial pressure of oxygen (pO_2_). Furthermore, actual bicarbonate (HCO3-), total concentration of carbon dioxide (tCO_2_), oxygen saturation (sO_2_), ionised calcium (Ca^2+^), packed cell volume (PCV), and haemoglobin (Hb) were measured. Base excess (BE) was calculated by each analyser. Chloride (Cl) was only measured with the Rapidlab (RL). While the RL measures via ion-selective electrodes, and only haematocrit and haemoglobin involve photometric measurements, the Osmetech (OSME) is based on a luminescence measurement, with only pCO_2_ being measured via ion-selective electrodes. The venous blood samples were cooled on ice and immediately transferred into the laboratory. The samples were centrifuged within 10 min and further analysed for total calcium (Ca), L-lactate, and D-lactate (Uvicon XL^®^, BioTeck Instruments GmbH, Bad Friedrichshall, Germany). Protein, albumin, aspartate aminotransferase (ASAT), glutamate dehydrogenase (GLDH), creatinine (crea), beta-hydroxybutyrate (β-HB), glucose (Gluc), magnesium (Mg), and phosphate (Phos) were analysed with Cobas Mira Plus^®^ (Roche Pharma AG, Basel, Switzerland) sodium (Na) and potassium (K) with flame photometry (Model 420, Sherwood Scientific Ltd., Cambridge, UK). Ruminal fluid and urine samples (Eppendorf Tubes^®^ 3810X, Eppendorf Vertrieb Deutschland GmbH, Wesseling, Germany) were cooled and transferred to the laboratory for analysis. Ruminal fluid was analysed for total calcium (ru_Ca), phosphate (ru_Phos) (Uvicon XL^®^), sodium (ru_Na), and potassium (ru_K) (Flammenfotometer Modell 420^®^), D- and L-lactate (Uvicon XL^®^). Urine was analysed for total calcium (u_Ca), phosphate (u_Phos) (Uvicon XL^®^), sodium (u_Na), potassium (u_K) (Flammenfotometer Modell 420^®^), glucose (u_gluc), creatinine (u_crea), and gamma-glutamyltransferase (u_GGT) (Cobas Mira Plus^®^). Kidney function tests were performed using paired blood and urine samples and values for creatinine clearance (Crea_cl), and fractional excretion rates (FE%) for water, sodium, phosphate, calcium, and potassium were calculated [15]. The sampling was performed according to the protocols of previous studies [12].

### 2.6. Analysis of Data

The data were analysed with Excel 2016 (Microsoft, Redmond, WA, USA) and SAS Enterprise Guide 7.1 (SAS Institute, Inc., Cary, NC, USA) for normality interpreting Q-Q-Plot, Boxplot, and Shapiro–Wilk test. Data were expressed as median (interquartile range) or mean (±standard deviation of mean (SD)) as appropriate. The analysis of data was performed according to the protocols of previous studies [12]. A total of 44 blood samples, 36 samples of ruminal fluid, and 39 urine samples were used for analysis.

## 3. Results

### 3.1. General Data on Anaesthesia

A total of 14 ewes were operated for laparotomy within this project. The mean BW was 39.8 kg (33 kg to 87 kg). The mean duration of anaesthesia was 3 h 47 min (±1 h 32 min). The mean ketamine amount administered in total in animals receiving a TIVA was 46.5 mg/kg BW (±9.2 mg/kg BW). Mean electrolyte infusion rate was 5 (±2) ml/kg BW/h. HR and RR revealed no great changes for hours one to three; for detailed data see Table 2. PE had a 34% higher HR compared to NPE.

Median rectal body temperature in sheep in TIVA was 38.8 (±1.3) °C and in InhA (37.7 (±1.2) °C). 

### 3.2. Blood Gas Analysis

In animals under InhA, mean for Hb was 24% lower and PCV was 26% lower than in animals under TIVA. In all sheep receiving a TIVA, mean pO_2_ was below the reference range, while in the InhA group, mean pO_2_ was within the reference range [22]. The difference between mean pO_2_ in ewes under InhA (148(97.5–176.3) mmHg) and ewes under TIVA (59.7(36.0–80.8) mmHg) was 88.3 mmHg. Data from animals additionally measured 2–8 h post-surgery (n = 7) are presented in Figure 1a. pCO_2_ was 6.2 mmHg higher at the last hour of anaesthesia (40.8(39.8–47.5) mmHg) compared to the awake standing animal (34.6(33.3–39.5) mmHg). Data for animals additionally measured 2–8 h post-surgery (n = 7) are presented in Figure 1b. sO_2_ in TIVA was below the reference range (91.0(77.0–94.7)%). Only one animal in the TIVA group was excluded, because for all three measurements during surgery, the sO_2_ was below the measurable limit. In the awake, standing animal, it was 94%. In the InhA group, sO_2_ was within the reference range (97.5(92.8–99.2)%). sO_2_ values measured during surgery and additionally in the awake, standing animals are presented in Figure 2 (n = 6). pH_art was slightly higher in PE (7.466(7.360–7.506)) than in NPE (7.451(7.399–7.500)). BE was lower in PE (4.8(2.0–7.0)) than in NPE (6.0(4.0–7.1)). Cl values did not change over time or when comparing InhA to TIVA, a difference of 4% could be observed when comparing PE (99.5(97.5–101.5) mmol/L) to NPE (103.0(100.0–103.0) mmol/L). Detailed data of blood gas analysis are in Appendix A. 

### 3.3. Blood Analysis in the Laboratory

Total protein revealed a decrease when comparing hour one (63.0(61.0–65.6) g/L) to hour two (56.5(55.3–62.0) g/L) and then remaining consistent to hour three (56.7(55.3–60.4) g/L). Albumin showed a decrease over time comparing hour one (28.1(26.0–30.0) g/L), two (26.6(24.4–27.4) g/L), and three (25.8(23.6–26.8) g/L). Regarding pregnancy status and type of anaesthesia, total protein and albumin concentrations displayed no great changes. Plasma glucose concentration in the blood increased from hour one (13.10(±4.58) mmol/L) to hour two (18.56 (±4.79) mmol/L) and hour three (21.72(±5.00) mmol/L). Differences in plasma glucose could also be detected when comparing InhA (15.08(±4.74) mmo/L) to TIVA (19.78(±6.04) mmol/L). Regarding Phos concentrations, a difference could be observed between InhA (1.24(1.05–1.49) mmol/L) and TIVA (1.51(1.16–2.04) mmol/L), the same as for Na regarding InhA (136.7(132.4–138.4) mmol/L) and TIVA (135.7(131.2–138.8) mmol/L). Mg concentrations were higher in PE (0.80(0.77-0.84) mmol/L) than in NPE (0.71(0.68–0.84) mmol/L). In ASAT, GLDH, Ca_tot, K, β-HB L-Lactate, and D-Lactate concentrations, hardly any differences could be detected over time or comparing InhA to TIVA or PE to NPE. Detailed data are presented in Appendix A.

### 3.4. Parameters in Ruminal Fluid

As for one animal, only one sample of ruminal fluid could be obtained, and the animal was excluded from the following analysis, which thereby contained data of 13 animals. Ru_Ca showed slight changes when comparing InhA (1.06(0.81–1.36) mmol/L) to TIVA (0.97(0.86–1.33) mmol/L) and a visibly higher concentration in PE (1.14(0.99–1.88) mmol/L) was evident compared to NPE (0.92(0.64–1.31) mmol/L); (Figure 3a). Ru_Phos displayed lower concentrations under InhA (17.0 (14.4–24.3) mmol/L) compared to TIVA (28.9(20.1–30.9) mmol/L) and a lower concentration in PE (14.5(13.8–17.4) mmol/L) compared to NPE (26.9(20.5–30.6) mmol/L); (Figure 3b). No remarkable difference was detected for ru_K, D-Lactate, L-Lactate, and ru_Na over time or when comparing InhA to TIVA or PE to NPE. Data are displayed in detail in Table 3. 

### 3.5. Parameters in Urine

Crea_cl was higher in sheep under InhA (1.79(1.56–2.07) mL/min/kg) compared to TIVA (1.54(1.39–1.67) mL/min/kg); very little difference was displayed over time or when comparing PE to NPE. Although in u_gluc no difference could be seen between InhA and TIVA or PE to NPE, it showed a clear increase from hour one (7.92(0.67–59.85) mmol/L) to two (111.87(42.11–220.83) mmol/L) and to hour three (223.1(103.2–328.54) mmol/L). Fractional excretion (FE%) for Ca, Phos, Na, K, and water and u_GGT presented no greater changes over time, nor when comparing InhA to TIVA or PE to NPE. Kidney function tests are presented in Table 4. 

## 4. Discussion

### 4.1. General Data on Anaesthesia

Although the overall number of sampled animals is limited, these data give an overview of sheep undergoing a laparotomy. The ewes were under general anaesthesia for approximately four hours and placed in the supine position. A control blood sample was taken from seven animals after awakening from anaesthesia. The relatively long operation time for the laparotomy was partly due to the surgery performed as part of the regular clinical training of the veterinary students. Therefore, slight differences in performing surgery, and likewise for obtaining the physiological parameters, have to be taken into account. A TIVA can be performed comparable to field conditions but it can lead to a less deep and more frequent breathing pattern. Although with InhA, mechanical ventilation with PEEP is possible, leading to better unfolding of the lungs with a more efficient gas exchange, it is more likely to be performed in a clinical setting as it is dependent on the presence of equipment, oxygen, and compressed air supply. That is why the respiratory rates were not comparable within the groups, as the sheep under InhA were for most of the time mechanically ventilated and thus had a very constant respiratory rate, while sheep under TIVA breathed spontaneously, and consequently, higher respiratory rates could be determined. Special attention is paid to the HR, though it is not a pain-specific parameter, it can be evaluated as an indicator of painful procedures [23]. Although HR in PE was higher, values in both groups were within the physiological range and showed no differences when comparing anaesthesia regime or time, leading to the assumption that the lumbosacral anaesthesia is sufficient to eliminate pain during this procedure. The median body temperature in the InhA group was slightly below the reference value [24] and 1.1 °C lower than under spontaneous breathing (TIVA group) where the maintenance of body temperature was probably easier due to less deep anaesthesia and less muscle relaxation. 

### 4.2. Blood Gas Analysis

Interestingly, we see clear differences in Hb and PCV between the anaesthesia groups, with more than 20% lower values in InhA; similar findings have already been described for pregnant sheep undergoing anaesthesia with acepromazine and isoflurane [25]. The authors project the changes to be the effect of an acepromazine-mediated splenic relaxation as described in horses and humans. Indeed, a similar reaction is described for dogs when receiving ketamine, because in non-splenectomised dogs, haemoglobin and packed cell volume were significantly lower than for splenectomised dogs [26]. In the present experiment, the animals did not receive acepromazine. The effect could only be seen in ewes under InhA, although all ewes received ketamine. Hence, we suspect the effect could be modulated by isoflurane due to deeper anaesthesia and increased muscle relaxation. Although the pregnancy status does not seem to have a greater impact on the pO_2_ and sO_2_, the form of anaesthesia did. In fact, in some samples, pO_2_ and sO_2_ in the TIVA group could not be measured, as these were below the detectable range for the analyser. Measurable values for pO_2_ still were more than 50% lower in the TIVA group compared to ewes under InhA. Similar effects could be observed in previous studies where ewes underwent surgery for Caesarean section without additional intubation and oxygen [10]. We consider that the supine position should be avoided in sheep, especially when oxygen supply of lambs is at risk. The administered oxygen and positive end-expiratory pressure improved oxygen supply in the ewes. A significant improvement in oxygenation of the ewes during emergency *SC*, due to oxygen delivered via a face mask, was also found by Musk et al. [11]. At hour three during surgery, the pCO_2_ was 6.2 mmHg higher than in the awake, standing animals after surgery. It becomes clear that the expiration of CO_2_ during anaesthesia is insufficient but normalises after returning to a stable standing position and walking after surgery. The CO_2_ elimination and especially the oxygen saturation are heavily influenced. In the TIVA group, three animals presented sO_2_ values below the detectable range during anaesthesia and returned to physiological saturation after stable standing and walking. The supine position might be difficult, as the rumen and full uterus exert significant pressure on the lungs, especially without PEEP. It is unclear whether the influence is due to atelectasis or shunts, as it can be experimentally induced by low pressure and volume [27]. The animals ventilated with PEEP and additional oxygen showed pCO_2_ values within the reference range. PE displayed 4% lower Cl values than NPE. The deep anaesthesia and the additional space needed by the uterus may lead to an increased reflux of Cl from the abomasum to the forestomach system and the missing Cl/HCO_3_^-^ exchange in the small intestine could contribute to alkalisation of the blood [28,29]. The differences in pH_art between PE and NPE remain minimal. In this study, Cl in ruminal fluid was not measured.

### 4.3. Blood Analysis in the Laboratory

The decrease seen in protein and albumin is likely to be a product of haemodilution due to infusion. An additional effect through activated coagulation due to minor bleeding during surgery cannot be excluded. A significant decrease in protein could also be seen in pregnant ewes under anaesthesia for laparotomy by Musk et al. [25]. The elevation in blood glucose together with the excretion of glucose in urine indicates an infusion of glucose-containing solutions (in this case 1000mL containing 55g glucose-monohydrate) exceeded the threshold of withholding glucose in the kidneys. It should be considered that high infusion of glucose could trigger a rebound effect in lambs, as is described in humans [30], or might interfere with the ewes’ glucose metabolism, as it is described that glucose infusion post-partum leads to lower food intake in ewes [31]. A full electrolyte solution containing less or no glucose should be preferred, even in pregnant ewes. In previous studies, we found that when anaesthesia is performed without infusion of glucose, the glucose levels in the blood drop to a significantly lower level (hour four after initiation of anaesthesia), which leads to a significant increase in β-HB (hour five after initiation of anaesthesia) [12]. As the anaesthesia for Caesarean section usually does not exceed three hours, an addition of glucose is unnecessary. The animals in the InhA group were intubated and unable to swallow. The saliva of sheep contains sodium as well as great amounts of phosphate, which physiologically is swallowed and reabsorbed in the gastrointestinal tract [8]. As the loss of saliva can be up to 700mL during eight hours of anaesthesia under intubation (unpublished data), we suspect the differences in Phos and Na to be due to loss of saliva.

### 4.4. Parameter in Ruminal Fluid

In PE, a higher ru_Ca concentration was detected in the ruminal fluid, while ru_Phos was lower than in NPE. The great amounts of phosphate contained in the saliva of sheep [8] is physiologically reabsorbed in the large intestine [32]. As most PE were under InhA and the intubation disables the animals to swallow saliva, we suspect the finding to be a statistical bias due to the study design. Unfortunately, ruminal changes during anaesthesia and fastening in sheep are so far not a main focus of research. Therefore, further investigations are needed, especially regarding the great influence of the rumen on electrolyte balance in sheep [13]. The consistent values of D-Lactate indicate that bacterial overgrowth does not seem to be a problem during anaesthesia, as mentioned in previous studies [12]. Whether an acidification took place could not be determined, but it is recommended to include ruminal pH measurements in further studies.

### 4.5. Parameters in Urine

A visible difference occurred in Crea_cl, with a higher rate in InhA compared to TIVA. This finding is contrary to results of previous studies, which described a reduction in renal blood flow to a higher extent in inhalational anaesthesia compared to totally intravenously anaesthetised sheep [33]. Still, as the median of all parameters of the kidney function test were within the reference range [15], we conclude that the infusion regime guaranteed a sufficient renal blood flow, although the perfusion rate was at the lower end of the recommended drip rate [34].

## 5. Conclusions

Based on the results, we can conclude that surgery in sheep placed in supine position should be ideally performed under inhalational anaesthesia, as pO_2_ and sO_2_ were higher when ventilated with PEEP and additional oxygen. Lower concentrations for Hb and PCV were found in animals operated under InhA. Changes in electrolytes in ruminal fluid occurred, especially between PE and NPE. Yet, too few data have been published to reliably evaluate the findings. As the rumen is of great importance for water and electrolyte status in sheep, this topic should be the focus of further research and we recommend including ruminal pH measurements. Kidney function could be maintained with a minimal electrolyte infusion regime and additional glucose is not mandatory, even in pregnant ewes.

## Figures and Tables

**Figure 1 animals-12-00834-f001:**
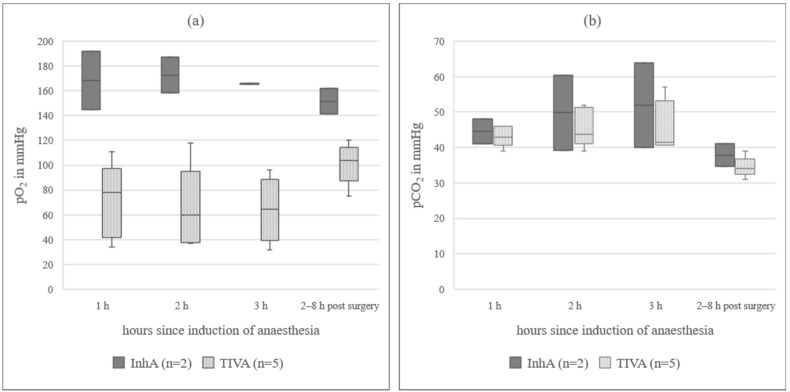
(**a**) **pO_2_ in arterial blood in sheep during three hours of anaesthesia**; InhA = inhalational anaesthesia, TIVA = total intravenous anaesthesia; the boxplots represent the interquartile range and the respective maximum and minimum, together with the median. (**b**) **pCO_2_ in arterial blood in sheep during three hours of anaesthesia**; InhA = inhalational anaesthesia, TIVA = total intravenous anaesthesia; the boxplots represent the interquartile range and the respective maximum and minimum, together with the median.

**Figure 2 animals-12-00834-f002:**
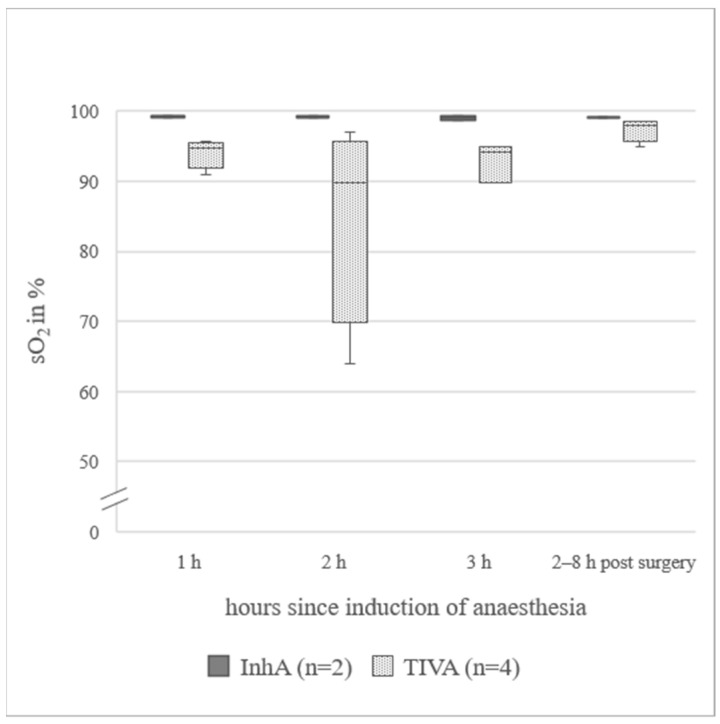
**sO_2_ in arterial blood of six sheep during surgery for laparotomy and afterwards in the awake, standing animal**; axis cropped at 50 %, InhA = inhalational anaesthesia, TIVA = total intravenous anaesthesia, sO_2_ = oxygen saturation; the boxplots represent the interquartile range and the respective maximum and minimum, together with the median.

**Figure 3 animals-12-00834-f003:**
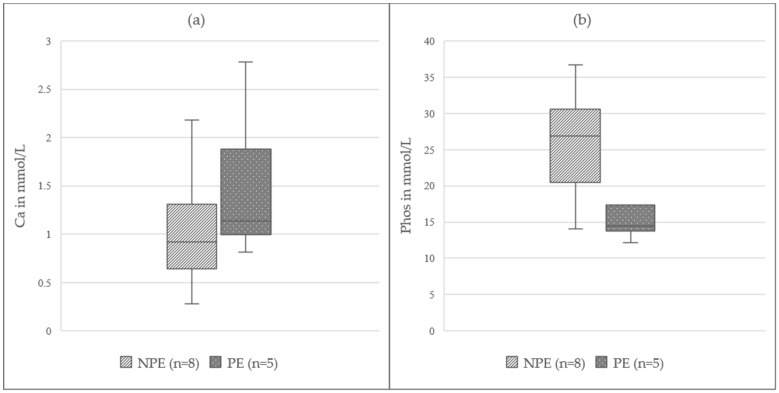
(**a**) **Total calcium in ruminal fluid of non-pregnant ewes (NPE) and pregnant ewes (PE) under general anaesthesia for laparotomy**; Ca = total calcium; the boxplots represent the interquartile range and the respective maximum and minimum, together with the median. (**b**) Phosphate in ruminal fluid of non-pregnant ewes (NPE) and pregnant ewes (PE) under general anaesthesia for laparotomy; Phos = phosphate; the boxplots represent the interquartile range and the respective maximum and minimum, together with the median.

**Table 1 animals-12-00834-t001:** Overview on anaesthesia in operated ewes.

Animal	Anaesthesia	Pregnancy status	Ketamine	Xylazine
No. 1	InhA	NPE	yes	yes
No. 2				
No. 3				
No. 4		PE		
No. 5				no
No. 6				
No. 7				
No. 8	TIVA	NPE	yes	yes
No. 9				
No. 10				
No. 11				
No. 12				
No. 13				
No. 14		PE		

InhA = inhalational anaesthesia, TIVA = total intravenous anaesthesia, NPE = non-pregnant ewe, PE = pregnant ewe.

**Table 2 animals-12-00834-t002:** Frequency of heartbeats and breaths per minute in sheep during anaesthesia for laparotomy.

	HR in Beats Per Minute		RR in Breaths Per Minute	
	TIVA	InhA	TIVA	InhA
hour 1	100(64–120)	88(76–96)	64(54–96)	14(12–53)
hour 2	64(48–100)	104(84–120)	60(44–64)	12(12–42)
hour 3	84(55–100)	112(76–118)	56(38–61)	12(12–40)

HR = heart rate, RR = respiratory rate, TIVA = total intravenous anaesthesia n = 7, InhA = inhalational anaesthesia n = 7; Data expressed as median (interquartile range).

**Table 3 animals-12-00834-t003:** Parameters in ruminal fluid of 13 sheep during laparotomy under two different anaesthesia protocols.

	InhA (n = 7)			TIVA (n = 6)		
Value	Hour 1	Hour 2	Hour 3	Hour 1	Hour 2	Hour 3
Sodium in mmol/L	110.2 ± 11	109.5 ± 15.3	107 ± 11.7	107.6 ± 8.5	109 ± 11.2	111.8 ± 4.3
Potassium in mmol/L	20.8(17.5–23.5)	22.9(18.4–26.4)	23.6(18.4–34.8)	23.3(21.5–27.2)	24.9(19.5–28.1)	23.7(18.2–36.7)
Calcium in mmol/L	0.9(0.8–1.1)	1.3(0.8–1.8)	1.3(0.8–2.1)	1(0.6–1.3)	1.1(0.8–1.6)	0.9(0.9–1.3)
Phosphate in mmol/L	17(14.1–20.5)	17.4(14.5–24.3)	16.3(13.7–26.9)	28.9(18.3–31.3)	26.1(18.7–31.1)	29.1(18.2–33.8)
L-Lactate in mmol/L	0.11(0.07–0.15)	0.11(0.06–0.13)	0.11(0.09–0.25)	0.11(0.08–0.17)	0.14(0.1–0.19)	0.14(0.12–0.22)
D-Lactate in mmol/L	0.1(0.09–0.13)	0.1(0.07–0.14)	0.12(0.11–0.17)	0.11(0.09–0.15)	0.13(0.11–0.19)	0.09(0.07–0.14)

InhA = Inhalational anaesthesia, TIVA = total intravenous anaesthesia; Data expressed as mean ± SD or median (Interquartile range).

**Table 4 animals-12-00834-t004:** Kidney function tests, gamma-glutamyltransferase, and glucose in the urine of sheep under general anaesthesia for laparotomy.

	InhA (n = 7)			TIVA (n = 7)			
Value	Hour 1	Hour 2	Hour 3	Hour 1	Hour 2	Hour 3	Reference Values *
Crea_cl in mL/min/kg	1.8(1.6–2.1)	1.7(1.6–2.1)	1.7(1.3–2)	1.6(1.4–1.7)	1.5(1.4–1.7)	1.5(1.3–1.6)	1.14–2.3
FE Calcium in %	0.8(0.1–1.6)	0.2(0.2–1.7)	0.7(0.1–1.6)	0.2(0.1–1.8)	0.8(0.1–1.6)	0.3(0–0.7)	0.03–8.9
FE Phosphate in %	0.2(0.1–2)	2.1(0.1–4.6)	0.2(0–2.4)	0.4(0.2–1.3)	1.9(0.2–3.0)	1.1(0.7–2.6)	0.01–3.39
FE Sodium in %	0.9(0–1.3)	0.9(0.2–1.8)	0.5(0.1–1.2)	0.4(0.2–1.1)	0.4(0.3–2.7)	0.6(0.3–1.8)	0.04–1.22
FE Potassium in %	56.9(46.1–72.1)	66.2(39.3–93.2)	33.1(16.7–52.8)	39.7(34.1–68)	74.3(37.7–104.1)	64.1(30.4–85.6)	12.2–96.0
FE Water in %	1.4(0.9–5.4)	1.5(0.7–7.6)	1.0(0.5–6.3)	2(0.8–4.5)	3.3(1.5–5.1)	2.8(1.9–4.1)	0.2–2.5
GGT in U/L	1.6(0.5–2.3)	3.3(0.7–3.5)	2.3(0.4–6.6)	1.4(0.8–2.8)	0.9(0.7–2.1)	1.3(0.9–2.7)	
Glucose in mmol/L	2.2(0.1–25.5)	86.8(25.6–293.9)	170.6(57.8–329.9)	17.3(9.4–81)	135.8(50.3–230.4)	244.5(142.4–328.5)	

InhA = Inhalational anaesthesia, TIVA = total intravenous anaesthesia, Crea_cl = creatinine clearance, FE = fractional excretion rate, GGT = gamma-glutamyltransferase; Data expressed as median (interquatile range), * Bickhardt K., Dungelhoef R. 1994.

## Data Availability

The data used and analyzed during the current study are available from the corresponding author on reasonable request.

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
