# Peer review of "Analysis of pH and Electrolytes in Blood and Electrolytes in Ruminal Fluid, including Kidney Function Tests, in Sheep Undergoing General Anaesthesia for Laparotomy"

_animals, 2022, doi:10.3390/ani12070834_

Round 1
Reviewer 1 Report
I revised the manuscript titled "Analysis of pH and electrolytes in blood and electrolytes in ruminal fluid, including kidney function tests in sheep undergoing general anesthesia for laparotomy" the overall idea is relevant. However, there are a lot of concerns regarding the statistical analysis conducted by the authors. I recommend a major revision of the manuscript before considering it for publication. Please see my comments below.
-Based on the description provided by the authors the experiment is a randomized design with a 2 x 2 factorial arrangement with two treatments ( InHA and TIVA) and four levels ( Pregnant and non-pregnant animals). Thus, the authors can estimate the effect of the type of Anaesthesia, pregnancy status, and their corresponding interactions.
-To consider the effects of time, the authors must account for the effects of beats per minute, breaths, blood measurements, and rumen fluid data should be analyzed as repeated measurements.
-The overall statistical model should account for the effects of Anaesthesia (InHA and TIVA) + pregnancy status (Pregnant and non-pregnant animals) + time (0, 1, 2, and 3 hours) and their interactions with the random effect of the animal.
L259 Please correct both the table and the graph. I was unable to read the table because of the graph.
Author Response
I revised the manuscript titled "Analysis of pH and electrolytes in blood and electrolytes in ruminal fluid, including kidney function tests in sheep undergoing general anesthesia for laparotomy" the overall idea is relevant. However, there are a lot of concerns regarding the statistical analysis conducted by the authors. I recommend a major revision of the manuscript before considering it for publication. Please see my comments below.
-Based on the description provided by the authors the experiment is a randomized design with a 2 x 2 factorial arrangement with two treatments ( InHA and TIVA) and four levels ( Pregnant and non-pregnant animals). Thus, the authors can estimate the effect of the type of Anaesthesia, pregnancy status, and their corresponding interactions.
Dear reviewer, thank you very much for your feedback and the important suggestions. We see, that the statistical model is inefficient and the low number of animals suggests a low test power. As the second reviewer annotated comparable points, we followed his suggestion and changed to a descriptive presentation of the data. We hope that the script now matches your expectations and will still visualize the important findings.
-To consider the effects of time, the authors must account for the effects of beats per minute, breaths, blood measurements, and rumen fluid data should be analyzed as repeated measurements.
See above.
-The overall statistical model should account for the effects of Anaesthesia (InHA and TIVA) + pregnancy status (Pregnant and non-pregnant animals) + time (0, 1, 2, and 3 hours) and their interactions with the random effect of the animal.
See above.
L259 Please correct both the table and the graph. I was unable to read the table because of the graph.
Thank you for the hint, we changed the layout, hopefully you can now read the table unrestricted. L271
Reviewer 2 Report
The paper focuses on influence of different anaesthesia regimes on electrolytes and acid-base-status in blood, urine and rumen liquid. It provides first data regarding this issue and the topic fits the scope of the journal. The writing is clear and readable.
However, there are some points that need revision.
Main point:
The statistics seem to be inappropriate:
There are two variable parameters (anaesthesia regime and pregnancy status) which are investigated in the same animals, i.e., which are not independent. Therefore, a two-way-analyses is necessary to compare these data. A separate comparison PE vs NPE or TIVA vs InhA is not valid. However, there is only one animal in the TIVA/PE group putting a statistical evaluation into question, at all. In general, the number of n is low suggesting a low test power. Hence, statistics should be interpreted with care (and test power should be provided additionally to p-values). Therefore, a descriptive presentation of the data may be more adequate.
The statements regarding the n number are confusing. Following your MM section, there are 7 ewes in the InhA group (3 NPE, 4 PE) and 7 ewes in the TIVA group (6 NPE, 1 PE). None of these numbers fits to the figures 1 or 2, even when regarding the exclusion of animals as stated in line 214 . Where are the lacking data? If n=2 is correct, a statistical analyses makes no sense.
Minor points:
Figures should be selfexplaining. Since symbols in a box plot can have different meanings, please provide information regarding the meaning of the crosses, lines and whiskers.
Table 3: data expressed as mean or median. What's correct?
Please, provide n for table 4.
line 309: there should be a space between "the" and "pregnancy status"
Line 310 ff: could the rumen catheder additionally have reduced the respiratory efficiency in the TIVA group?
line 330 ff: wouldn't a higher Cl concentration in the forestomach activate Cl-/HCO3- exchangers in the ruminal wall and, therefore acidify the blood?
Author Response
The paper focuses on influence of different anaesthesia regimes on electrolytes and acid-base-status in blood, urine and rumen liquid. It provides first data regarding this issue and the topic fits the scope of the journal. The writing is clear and readable.
However, there are some points that need revision.
Main point:
The statistics seem to be inappropriate:
There are two variable parameters (anaesthesia regime and pregnancy status) which are investigated in the same animals, i.e., which are not independent. Therefore, a two-way-analyses is necessary to compare these data. A separate comparison PE vs NPE or TIVA vs InhA is not valid. However, there is only one animal in the TIVA/PE group putting a statistical evaluation into question, at all. In general, the number of n is low suggesting a low test power. Hence, statistics should be interpreted with care (and test power should be provided additionally to p-values). Therefore, a descriptive presentation of the data may be more adequate.
Dear Reviewer, than you very much for your feedback! We see as well, that due to the low number of animals the test power is low. As the first reviewer annotated similar points on the statistics, we followed your suggestion and changed the presentation of the data to a descriptive form. We hope that the script will match your expectations and will still visualize the important findings. Please see “Results” L 198 ff.
The statements regarding the n number are confusing. Following your MM section, there are 7 ewes in the InhA group (3 NPE, 4 PE) and 7 ewes in the TIVA group (6 NPE, 1 PE). None of these numbers fits to the figures 1 or 2, even when regarding the exclusion of animals as stated in line 214 . Where are the lacking data? If n=2 is correct, a statistical analyses makes no sense.
Thank you very much for the attentive comments. In the script, the declaration of Figure 1a and b had a problem with the visualisation. We changed the Figures accordingly. For pO2 and pCO2 (Figure 1) data of seven animals were included, for sO2 (Figure 2) six animals were included, as one animal was excluded. It should now be clear to the reader. L 230; L 238
Minor points:
Figures should be selfexplaining. Since symbols in a box plot can have different meanings, please provide information regarding the meaning of the crosses, lines and whiskers.
Thank you for the suggestions. We agree, and changed the descriptions accordingly. L232 ff; L240 ff; L277 ff
Table 3: data expressed as mean or median. What's correct?
Thank you for the question, according to our colleagues in the statistic department, it is appropriate to express normally distributed data with the mean (±standard deviation) and non-normally distributed data with median [interquartile range]. That is why we chose the mixed representation.
Please, provide n for table 4.
We appreciate the comment, and changed the description accordingly. L293 ff.
line 309: there should be a space between "the" and "pregnancy status"
Thank you for the hint, we added a space accordingly. L 330
Line 310 ff: could the rumen catheder additionally have reduced the respiratory efficiency in the TIVA group?
We appreciate the comment on this interesting aspect. Even if it cannot be completely ruled out; we do not assume such an influence. In fact, we place the ruminal tubes also during routine surgeries to lower ruminal pressure on the lungs (less gas accumulation) in order to contribute to improvement of breathing. We use these (0.6 cm diameter) tubes also in the awake animals (e.g to administer liquids to the rumen), and we do not see any hints for restriction in the respiration.
line 330 ff: wouldn't a higher Cl concentration in the forestomach activate Cl-/HCO3- exchangers in the ruminal wall and, therefore acidify the blood?
Thank you very much for this important hint. We realized a descriptive mistake, on our side, and changed this accordingly. L 335 ff.
Still, we suspect a reflux of abomasal ingesta to contribute to a hypochloraemic alkalosis (as e.g. after obstruction/abomasal displacement etc.) as the chloride is missing in the small intestine for exchange of Cl for HCO3-. Though the mentioned transport system clearly exists in the ruminal wall, clinical experience suggests that the transporter in the rumen cannot compensate the missing transport in the jejunum. For easier reading, we added this information in the text. L 352 ff
Reviewer 3 Report
The surgical management of parturition in ruminants is an event that has challenges due to the anatomical characteristics of the animals, as described by the authors. To date, the different anesthetic protocols, such as those used in this study (TIVA or inhalational anesthesia), have advantages or disadvantages that can be reflected in biochemical parameters. This study aims to provide relevant information on this aspect. However, I have left some comments and suggestions since one of the weaknesses of this study is the lack of connection between the title, aim and conclusion.
Line 2: The title and the aim of the study are not connected. I would suggest rewriting the title or the objective of the work. For example, the title could be changed to: Comparison of blood parameters, ruminal fluid, and renal function in ewes under two protocols of general anesthesia for laparotomy”. Or the aim could be changed to: “to compare/evaluate the effects of injectable anesthesia and inhalational anesthesia on blood parameters, ruminal fluid and kidney function of ewes undergoing laparotomy”
Line 25: Please, add a space between “where” and “of”.
Lines 33-34: Please, clearly describe which significant differences were observed in the subjects.
Line 39: Considering the content of the manuscript, authors could include “injectable anesthesia” and “inhalational anesthesia” as keywords
Line 45: In the introduction, I think it needs to be mentioned that laparotomy in ewes can be performed with inhalational and TIVA anesthesia. Authors could discuss some of the current knowledge about the advantages/disadvantages of both, so the reader could understand why authors are comparing perioperative parameters during both anesthesia protocols.
Lines 48-49: Mentioning NSAIDs sounds a little confusing since the aim of the study was not assessing this analgesic drug. I would suggest rewriting the idea.
Line 51: Add a space after “(44.3%)”
Line 56: I suggest starting a new paragraph from “There are different techniques”. In general, the Introduction is solely one long paragraph.
Lines 75-78: I would suggest restructuring the aim since one of the main points of this study was comparing TIVA and inhalational anesthesia.
Line 81: In this section I think authors need to include more information regarding the exclusion and inclusion criteria for the 14 animals. Details such as the breed, the selected age, if the animals were clinically healthy before the surgical procedure, and, in general, demographical information of the ewes included in the study.
Line 89: I suggest including in these lines the criteria the authors used to select the animals in each group. It would be adequate to also include the type of study (prospective, retrospective), and if it was a blinded study.
Line 91: Please, include if the animals received fluid therapy, since this could influence your results. Likewise, please, include the vein where xylazine was administered and if the authors evaluated the depth of sedation before anesthesia induction.
Line 96: Please, include which criteria the authors used to evaluate the surgical plane or unconsciousness of animals.
Lines 98-99: Maybe what the authors mean is that “Anaesthesia was then maintained with isoflurane at a minimal alveolar concentration of 0.5-2.5%”?
Lines 101-103: Include if the ventilator was the one included in the anesthesia machine or a separate one, including the brand of the ventilator.
Lines 105-109: Please, clarify if the quantity of administered ketamine and xylazine was calculated per hour or per minute.
Line 110: Cleary state the infusion velocity and the blood vessel where fluid therapy was administered.
Line 111: I would suggest including the NSAID that was administered and the dose (if it was a single dose, etc.).
Line 112: Regarding the use of procaine, it is not clear whether authors used it via epidural in both experimental groups. If that was not the case, state so and explain why the protocols differ.
Line 117: Table 1 is not cited in the text.
Line 190-194: This information could be included in the methods section instead of results.
Line 196: I recommend writing the “p” of the p-value in lowercase and italics throughout the manuscript.
Lines 201-202: I suggest presenting the temperature as: 38.8 ± 1.3°C instead of 38.8 (±1.3) °C.
Lines 210-213: Do not forget to put the data obtained not only p-value when you describe results. For example, pCO2 was also significantly higher (value) at hour three of anesthesia compared to the awake standing animal (value)(p=0.0289), while no statistical difference could be detected when comparing InhA to TIVA (p-value) and PE to 212 NPE (p-value).
Line 213: Please, specify the value of sO2 that was below the reference range.
Lines 216-217: I recommend including the parameters of sO2 in InhA and TIVA patients, not only a p-value
Line 226-228: Please, include a title for Figure 1. (a). Also, correct subscript (e.g., pO2)
Line 227: Please, correct subscript (e.g., pCO2)
Line 230-232: Include a title for the Figure.
Line 258: I suggest changing the title of the table to Mean and SD of parameters in the ruminal fluid of 13 sheep during two types of anesthesia for laparotomy
Table 3 and Figure 3 (a) (b) are overlapped. Also, provide a title for Figure 3.
Line 280: I would suggest starting the discussion by highlighting the findings of the study regarding the differences between TIVA and InhA subjects.
Lines 287-288: While HR is one of the parameters commonly used to assess surgical pain, it is not considered a pain-specific parameter. This information could be briefly mentioned.
Line 291-294: I suggest discussing further about respiratory rate, referring to the advantages/disadvantages of TIVA and inhalational anesthesia on the respiratory pattern. Likewise, the effect of fluid therapy to modulate tissular perfusion needs to be discussed.
Line 294-298: The reference value is the basal parameter of the ewes? Please, clarify. Also, the authors could discuss the physiological basis of body temperature and vasodilatation using inhalational anesthesia.
Line 309: Add a space between “the” and “pregnancy”
Lines 341-343: Try to include some references in veterinary medicine
Line 344: Add a reference to back up this statement, please.
Lines 365-366: I would suggest including these lines in the conclusion.
Line 375: Including the limitations of the study would be appropriate. For example, the authors stated that veterinary students performed the surgeries, and research has shown that the skill of the surgeon can influence the degree of surgical stress and, therefore, biochemical parameters. Other examples are the person(s) who evaluated the physiological parameters, or if the blood pressure measurement was invasive or non-invasive.
Author Response
The surgical management of parturition in ruminants is an event that has challenges due to the anatomical characteristics of the animals, as described by the authors. To date, the different anesthetic protocols, such as those used in this study (TIVA or inhalational anesthesia), have advantages or disadvantages that can be reflected in biochemical parameters. This study aims to provide relevant information on this aspect. However, I have left some comments and suggestions since one of the weaknesses of this study is the lack of connection between the title, aim and conclusion.
Line 2: The title and the aim of the study are not connected. I would suggest rewriting the title or the objective of the work. For example, the title could be changed to: Comparison of blood parameters, ruminal fluid, and renal function in ewes under two protocols of general anesthesia for laparotomy”. Or the aim could be changed to: “to compare/evaluate the effects of injectable anesthesia and inhalational anesthesia on blood parameters, ruminal fluid and kidney function of ewes undergoing laparotomy”
Thank you for your feedback! We extended the aims of our study to include the different anaesthesia protocols. We would stay with the title it is one of the first documentation of such parameters in sheep during anaesthesia in general. L 83
Line 25: Please, add a space between “where” and “of”.
We ment “whereof” (as “from which” or “of which”).
Lines 33-34: Please, clearly describe which significant differences were observed in the subjects.
Thank you for the hint. We added further details. L 30ff.
Line 39: Considering the content of the manuscript, authors could include “injectable anesthesia” and “inhalational anesthesia” as keywords
Thank you very much for the very good suggestion, we added keywords to the list. L 42
Line 45: In the introduction, I think it needs to be mentioned that laparotomy in ewes can be performed with inhalational and TIVA anesthesia. Authors could discuss some of the current knowledge about the advantages/disadvantages of both, so the reader could understand why authors are comparing perioperative parameters during both anesthesia protocols.
Thank you for the hint we added further information. L71 ff
Lines 48-49: Mentioning NSAIDs sounds a little confusing since the aim of the study was not assessing this analgesic drug. I would suggest rewriting the idea.
We appreciate your thought on this, we only mentioned the administration of NSAIDs for completeness, as in Germany it remains a problem finding licenced pharmaceuticals associated with anaesthesia or pain relief.
Line 51: Add a space after “(44.3%)”
Thank you very much, we corrected the mistake. L 52
Line 56: I suggest starting a new paragraph from “There are different techniques”. In general, the Introduction is solely one long paragraph.
We appreciate the comment, and separated the introduction to three paragraphs for easier reading. L60 ff.
Lines 75-78: I would suggest restructuring the aim since one of the main points of this study was comparing TIVA and inhalational anesthesia.
We agree with this point, and added the anaesthesia protocols to the aim. L 83
Line 81: In this section I think authors need to include more information regarding the exclusion and inclusion criteria for the 14 animals. Details such as the breed, the selected age, if the animals were clinically healthy before the surgical procedure, and, in general, demographical information of the ewes included in the study.
Thank you for the suggestion, we added further details. L 87f; L91 ff
Line 89: I suggest including in these lines the criteria the authors used to select the animals in each group. It would be adequate to also include the type of study (prospective, retrospective), and if it was a blinded study.
We agree and added more details to the material and methods section L91. The study was not blinded and operations and data collection took place in regular teaching in the clinic. We added Information about limitations of the study. L 296; L301 f . The laboratory personnel were not informed about the form of anaesthesia or pregnancy status.
Line 91: Please, include if the animals received fluid therapy, since this could influence your results. Likewise, please, include the vein where xylazine was administered and if the authors evaluated the depth of sedation before anesthesia induction.
Thank you for the hints, all animals received intravenous fluids; we created a paragraph and changed the description to further clarify. L 120 ff. xylazine was not administered intravenously, but i.m. L 115 and we did not further evaluate the depth of sedation, but for the intubated animals, the absence of swallowing was necessary.
Line 96: Please, include which criteria the authors used to evaluate the surgical plane or unconsciousness of animals.
Thank you for the comment, we have a minimal evaluation during surgery, which includes pupillary reflex, movement or swallowing and heart and respiratory rate and is continuously observed by the responsible veterinarian.
Lines 98-99: Maybe what the authors mean is that “Anaesthesia was then maintained with isoflurane at a minimal alveolar concentration of 0.5-2.5%”?
As minimal alveolar concentration is a defined concentration of an aesthetic agent that prevents motoric response from 50% of the animals it is more a value to compare potency of anaesthetic agents. What we meant here was simply the % of anaesthetic in oxygen in inspirated air.
Lines 101-103: Include if the ventilator was the one included in the anesthesia machine or a separate one, including the brand of the ventilator.
The ventilator is part of the semi-closed circuit rebreathing system (Modell CATO®, Dräger Medizintechnik GmbH, Lübeck, Germany).
Lines 105-109: Please, clarify if the quantity of administered ketamine and xylazine was calculated per hour or per minute.
We appreciate this important hint, it was bolus administration of half the initial dose of ketamine after 10 to 30 min, we now added further details. L 116f
Line 110: Cleary state the infusion velocity and the blood vessel where fluid therapy was administered.
Thank you for the hint, the velocity is further mentioned in the results section 3.1. L203 but we modified the sentence to clarify. The blood vessel is mentioned under 2.2. preparation of the animal L133.
Line 111: I would suggest including the NSAID that was administered and the dose (if it was a single dose, etc.).
Thank you, we can follow the idea. The ewes received different NSAIDs, and as you mentioned before, pain elimination is not main aim of this research we tried to ease the reading of the main manuscript.
Line 112: Regarding the use of procaine, it is not clear whether authors used it via epidural in both experimental groups. If that was not the case, state so and explain why the protocols differ.
Thank you very much for the important hint, we now created an extra paragraph to claify, that all 14 ewes received procaine. L 120 ff.
Line 117: Table 1 is not cited in the text.
We appreciate the hint and added a reference in the text. L 126 f
Line 190-194: This information could be included in the methods section instead of results.
Thank you for the suggestion. We now added further details to the material and methods section. L 87f
Line 196: I recommend writing the “p” of the p-value in lowercase and italics throughout the manuscript.
Reviewer one and two stated that due to the low number of animals the test power is low. We thereby followed the suggestion and changed the presentation of the data to a descriptive form. We hope that the script will match your expectations and will still visualize the important findings. Thereby no more p-values should appear in the text.
Lines 201-202: I suggest presenting the temperature as: 38.8 ± 1.3°C instead of 38.8 (±1.3) °C.
Thank you for the suggestion. This would as well be a good presentation; As we introduced the mean (±SD) presentation initially, we would like to stick to a continuous form of presentation in tables and text. L 194
Lines 210-213: Do not forget to put the data obtained not only p-value when you describe results. For example, pCO2 was also significantly higher (value) at hour three of anesthesia compared to the awake standing animal (value)(p=0.0289), while no statistical difference could be detected when comparing InhA to TIVA (p-value) and PE to 212 NPE (p-value).
We absolutely agree, as we anyhow changed major points in our presentation of results (descriptive form) we hopefully now as well covered all aspects of this comment. L 198 ff
Line 213: Please, specify the value of sO2 that was below the reference range.
We appreciate the comment, in the additional Table S1 we further express detailed values of sO2.
Lines 216-217: I recommend including the parameters of sO2 in InhA and TIVA patients, not only a p-value
See above.
Line 226-228: Please, include a title for Figure 1. (a). Also, correct subscript (e.g., pO2)
Thank you for the hint, we now highlighted the titles of tables and figures with bold letters, and corrected the subscript. L 232
Line 227: Please, correct subscript (e.g., pCO2)
Thank you, we changed the text accordingly. L 234
Line 230-232: Include a title for the Figure.
See above.
Line 258: I suggest changing the title of the table to Mean and SD of parameters in the ruminal fluid of 13 sheep during two types of anesthesia for laparotomy
Thank you very much, we added the anaesthesia protocols in the title as you suggested. L271
Table 3 and Figure 3 (a) (b) are overlapped. Also, provide a title for Figure 3.
We changed the layout, hopefully it is now visible.
Line 280: I would suggest starting the discussion by highlighting the findings of the study regarding the differences between TIVA and InhA subjects.
Thank you, this is a good idea, still we decided to present and discuss the findings in the same order for reasons of clarity.
Lines 287-288: While HR is one of the parameters commonly used to assess surgical pain, it is not considered a pain-specific parameter. This information could be briefly mentioned.
Thank you very much for this comment, we rephrased the sentences to clarify this information for the reader. L 311
Line 291-294: I suggest discussing further about respiratory rate, referring to the advantages/disadvantages of TIVA and inhalational anesthesia on the respiratory pattern. Likewise, the effect of fluid therapy to modulate tissular perfusion needs to be discussed.
Thank you for the suggestion, we added further details. L302ff
Line 294-298: The reference value is the basal parameter of the ewes? Please, clarify. Also, the authors could discuss the physiological basis of body temperature and vasodilatation using inhalational anesthesia.
Thank you, we added the citation for the reference value. L316
Line 309: Add a space between “the” and “pregnancy”
Thank you, we changed the spelling accordingly. L330
Lines 341-343: Try to include some references in veterinary medicine
We appreciate the comment and added further information. L 366
Line 344: Add a reference to back up this statement, please.
This is a statement made by us and corresponds to the very high glucose values measured in the blood and urine even in pregnant ewes.
Lines 365-366: I would suggest including these lines in the conclusion.
Thank you for the hint, we agree and added the lines to the conclusion. L 405 f
Line 375: Including the limitations of the study would be appropriate. For example, the authors stated that veterinary students performed the surgeries, and research has shown that the skill of the surgeon can influence the degree of surgical stress and, therefore, biochemical parameters. Other examples are the person(s) who evaluated the physiological parameters, or if the blood pressure measurement was invasive or non-invasive.
We appreciate the comment and added further information on the limitations. L 301 ff; L 296
Round 2
Reviewer 2 Report
Thank you for revising your manuscript. From my view, it clearly improves the quality.
Please, check again for some spelling/punctuation errors, especially in the inserted text.
Reviewer 3 Report
The authors have made an extraordinary effort to improve the manuscript, and they have followed all my recommendations. There have been many changes but all have contributed to improve their article.
The article is now ready for publication. I have no further comments.